# Age- and ApoE Genotype-Dependent Transcriptomic Responses to O_3_ in the Hippocampus of Mice

**DOI:** 10.3390/ijms26062407

**Published:** 2025-03-07

**Authors:** Mary F. Nakamya, Kaili Hu, Chunsun Jiang, Zechen Chong, Rui-Ming Liu

**Affiliations:** 1Department of Medicine, University of Alabama at Birmingham, Birmingham, AL 35294, USA; mfnakamya@uabmc.edu (M.F.N.); chunsunjiang@uabmc.edu (C.J.); 2Department of Biomedical Informatics and Data Science, Heersink School of Medicine, University of Alabama at Birmingham, Birmingham, AL 35294, USA; klhu0502@uab.edu

**Keywords:** aging, ozone, APOE, oxidative stress, synaptic function, neurogenesis, Alzheimer’s disease

## Abstract

Alzheimer’s disease (AD) is a leading cause of dementia in the elderly, with late-onset AD (LOAD) accounting for 95% of the cases. The etiology underlying LOAD, however, remains unclear. Using a humanized mouse model, we showed previously that exposure to ozone (O_3_), a potential environment risk factor, in a cyclic exposure protocol that mimics a human exposure scenario, accelerated AD-like neuropathophysiology in old humanized male ApoE3 (E3) but not ApoE4 (E4) mice. Using RNA sequencing (RNA-seq) techniques, we further demonstrate here that the ApoE genotype has the greatest influence on transcriptional changes, followed by age and O_3_ exposure. Notably, AD-related genes were expressed even at baseline and in young mice, but the differences in the expression levels are obvious in old age. Importantly, although both E3 and E4 mice exhibited some AD-related transcriptomic alterations, old E3 mice exposed to O_3_, which showed memory impairment, experienced more pronounced disruptions in the expression of genes related to redox balance, neurogenesis, neuroinflammation, and cellular senescence in the hippocampus, compared with O_3_-exposed old E4 mice. These results provide new insights into the molecular mechanisms underlying memory loss in O_3_-exposed old E3 male mice and emphasize the complexity of interactions between gene, environment, and aging in AD pathophysiology.

## 1. Introduction

Alzheimer’s disease is a progressive neurodegenerative disorder characterized by cognitive decline, memory loss, and behavioral changes, making it the leading cause of dementia in the elderly worldwide. In the United States alone, approximately 6.7 million individuals aged 65 and older are currently living with AD, and this number is expected to increase as the population ages [1,2]. Despite extensive research over several decades, the precise causes of AD remain elusive. Accumulated evidence suggests that the disease is driven by a multifaceted interaction between genetic predispositions, environmental exposure, and aging [3]. Old age is the most significant contributor to the onset of AD, although the mechanism underlying aging-related susceptibility to AD is still not well understood. Apolipoproteins (apoE), which play a critical role in lipid metabolism, have three isoforms, coded by three alleles ε2, ε3, and ε4 [4,5]. ε4, carried by 10–15% of the human population, is believed to have the strongest link with AD among the identified genetic risk factors so far. Most of the human population (80–85%) carries ε3, whereas 2–3% of the population carries ε2, a protective allele. It is estimated that the ε3/ε4 allele increases the risk of AD by 3- to 4-fold, whereas the risk is increased 10- to 15-fold in ε4/ε4 carriers compared with ε3 carriers [6,7]. Although ε4 is believed to be the most significant genetic risk factor for AD, its presence alone, even in the elderly, is not sufficient to cause the disease. Other factors, in addition to old age and the *ApoE* ε4 gene, must contribute. Epidemiological studies show that exposure to environmental pollutants, including PM2.5 and O_3_, is associated with cognitive impairment in the elderly, especially when combined with genetic susceptibility [8,9,10]. Nonetheless, whether and how environmental pollutants interact with genetic risk factors and/or aging leading to LOAD remains to be determined.

Ozone is one of the most abundant urban pollutants. Over 30% of the US population lives in areas with unhealthy levels of O_3_ (American Lung Association State of the Air 2012). In occupational settings, particularly for people working in outdoor environments such as construction, agriculture, and transportation, prolonged exposure to elevated amounts of ground-level O_3_ is also a prevalent hazard [11,12,13]. This continued exposure raises significant concerns about the long-term neurological health of workers. O_3_ is a highly reactive oxidant. Studies, including our own, have shown that O_3_ exposure induces oxidative stress in the brain and impairs memory in mice and rats [10,14]. O_3_ exposure has also been shown to activate microglia and promote neuroinflammation via a lung–brain axis, leading to memory impairment and neurodegeneration [15]. Importantly, recent epidemiologic studies show a positive correlation between O_3_ levels and incidence of AD [16,17]. Additionally, O_3_ exposure disrupts blood–brain barrier (BBB) integrity, promoting Aβ plaque accumulation and dystrophic neurite formation, impairing synaptic function and accelerating cognitive decline [18]. The lung–brain axis is central to these effects as O_3_-induced pulmonary inflammation triggers systemic immune activation, enabling inflammatory mediators to cross the BBB and amplify neuroinflammation [19]. This mechanism highlights how inhaled pollutants accelerate neurodegeneration, particularly in aging individuals with weakened immune and vascular defenses. Nonetheless, whether O_3_ is a culprit for AD and whether O_3_ acts alone or synergizes with other risk factors, such as *Apoe ε4* and/or aging, leading to AD, remains to be determined.

Humanized E3 and E4 mouse models are extensively utilized to investigate the role of the genetic risk factor *ApoE ε4* in the pathophysiology of AD. In a previous study, we showed that the exposure of mice to a cyclic O_3_ exposure protocol, which mimic human exposure scenarios, impairs memory in old E3, but not old E4 or young E3 and E4, male mice. Given these findings, the current study aims to explore the transcriptomic changes induced by O_3_ in the hippocampus of these mice. By examining the interplay between age, genotype, and environmental exposure, this study seeks to provide insights into the molecular mechanisms driving the increased susceptibility of old E3 males to the neurotoxic effects of O_3_. Understanding these mechanisms is crucial for developing knowledge-based interventions for populations at risk, especially individuals with exposure to elevated O_3_ levels.

## 2. Results

### 2.1. Impact of ApoE Genotype, Aging, and Ozone Exposure on Gene Expression in the Hippocampus of Male Mice

To explore the molecular mechanism underlying the increased susceptibility of old male E3 mice to O_3_-induced memory loss, we employed RNAseq techniques to investigate the impact of ApoE genotype, aging, and O_3_ exposure on the hippocampal transcriptome of humanized E3 and E4 male mice. Differentially expressed genes (DEGs) were evaluated across three factors: genotype (E4 vs. E3), age [17 months (17M) vs. 3 months (3M)], and exposure [ozone (O_3_) vs. filtered air (FA)]. To validate the RNAseq findings, a subset of genes as listed in Appendix A were further analyzed using quantitative real-time polymerase chain reaction (qRT-PCR). The Venn diagram (Figure 1a) illustrates distinct patterns of DEGs, with 1229 genes being specifically upregulated in E4 compared with E3, 612 genes in 17M vs. 3M mice, and 179 genes in O_3_-exposed vs. FA groups. Interestingly, 17 genes were upregulated across all three factors, 115 genes (5%) upregulated in both E4 and 17M mice, 129 genes in both E4 and O_3_-exposed mice, while 16 genes (1%) were in both 17M and O_3_-exposed mice (Figure 1a). In contrast, 994 genes were downregulated specifically in E4 vs. E3, 360 genes were downregulated in 17M vs. 3M mice, and 195 genes were downregulated in the O_3_ vs. FA groups (Figure 1b). Meanwhile, 14 genes were downregulated in both E4 and 17M mice, three genes in both E4 and O_3_-exposed mice, while nine genes were in both 17M and O_3_-exposed mice. The overlaps highlight the intersecting effects of genotype, age, and treatment, while unique genes emphasize the distinct influence of each factor. A detailed list of all genes is provided in (Appendix A).

A functional analysis of the 17 upregulated intersection genes highlights their role in aging- and stress response-related processes. For example, *Pon3* and *Trpv4* are involved in oxidative stress and calcium signaling disruptions, *Enpp2* is involved in altered neural development, while. *Lbp* and *Wfdc2* are involved in inflammatory responses. The upregulation of these genes underscores the combined effects of aging, genotype, and O_3_ exposure. Several genes are upregulated across two risk factors. *Ifi44* (inflammation) and *Irf7* (immune regulation) are elevated in E4 and aging mice, while *Sod3* (antioxidant defense) and *Glp1r* (neuroprotection) are increased in E4- and O_3_-exposed mice. On the other hand, *Nr4a2* (dopaminergic neuron development) and *Cacna1g* (calcium channel regulation) are suppressed in aging and E4 mice, while *Gprin3* (neural signaling) is downregulated in both aging and O_3_ exposure. A full list of genes that changed with age, genotype, and O_3_ exposure can be found in Appendix A. These findings suggest that aging and O_3_ exposure contribute to AD pathology through genotype-dependent mechanisms, particularly via oxidative stress, immune activation, and disrupted calcium signaling.

Volcano plots (Figure 1c–e) show that *Ccdc159* (structural organization), *Tcea2* (transcription elongation), and *Pxdn* (oxidative stress regulation) are down regulated whereas *Fbxw11* (protein degradation) is upregulated in E4 mice, compared with E3 mice (Figure 1c). On the other hand, *Pcdhb9* (synapse formation) and *Lgals9* (immune modulation) are upregulated in old mice, while *Gpr17* (oligodendrocyte maturation) and *Selenbp2* (antioxidant defense) are downregulated in old vs. young mice, indicating vulnerabilities in neurodegeneration (Figure 1d). O_3_ exposure led to the upregulation of *Gpx8* (antioxidant) and *Sfrp5* (Wnt signaling antagonist), alongside the downregulation of *Kcne2* (potassium channel regulation) and *Krt7* (epithelial integrity) (Figure 1e). These changes point to oxidative stress, disrupted signaling, and cellular dysfunction as key contributors to O_3_-induced memory impairment in old E3 mice.

To further dissect the role of genotype, age, and O_3_ exposure on the gene expression, we compared E4 and E3, young and old, O_3_/FA-exposed mice using KEGG pathway and heat map analysis (Figure 2). In E4 mice, the major upregulated pathways included leukocyte transendothelial migration and neuroactive ligand–receptor interaction, while downregulated pathways involved neurotrophin signaling and the inflammatory mediator regulation of TRP channels. Notably, the neuroactive ligand–receptor interaction pathway showed both up- and downregulation, suggesting a compensatory response to oxidative stress in E4 mice (Figure 2a). In old mice, the major upregulated pathways involve PI3K-Akt signaling, AGE-RAGE signaling, and chemokine signaling, reflecting immune and metabolic dysregulation. Downregulated pathways, including neuroactive ligand–receptor interaction, cAMP signaling, and calcium signaling, indicate impaired neuronal signaling and synaptic function (Figure 2b). In O_3_-exposed mice, the downregulation of cAMP and neuroactive ligand–receptor pathways pointed to impaired neurotransmission, while upregulated Wnt signaling suggested a compensatory mechanism (Figure 2c). The list of genes for the KEGG pathway are shown in (Appendix A).

A helical heat map of the top 40 DEGs in E4 vs. E3 mice revealed the upregulation of *Pon1* and *Ttr* (Figure 2d). *Pon1* supports antioxidant defense, while *Ttr* aids in preventing amyloid aggregation, suggesting a protective mechanism against O_3_-induced oxidative stress in E4 mice. In old vs. young mice, immune- and inflammation-related genes such as *Ighg2b*, *Saa3*, *Tnf*, and *S100a5* were upregulated (Figure 2e), reflecting age-related immune activation and neuroinflammation, both hallmarks of AD (Figure 2e). While *Bcl2l10* (anti-apoptotic) was upregulated in O_3_-exposed mice, the top 40 DEGs lacked other AD-related genes (Figure 2e). These findings suggest genotype and aging are the primary drivers of AD pathology, with O_3_ exposure playing a secondary exacerbating role.

### 2.2. Specific Effect of ApoE Genotype, Age, or O_3_ Exposure on the Gene Expression in the Hippocampus of Mice

To investigate the specific effects of genotype, age, and O_3_ exposure on gene expression, we conducted a detailed subgroup analysis. This approach allowed us to examine how each factor independently influences molecular pathways and contributes to memory loss (AD). The results show that aging affects E3 and E4 mice differently, with E4 favoring neuroprotection and E3 showing functional decline (Figure 3a). In old E4 mice, *Gh*, *Ramp3*, *Cpne9*, *Pgd2*, and *Gabrd* were upregulated, supporting neuronal maintenance and neuroprotection, while *Met*, *Gpr161*, and *Dio3* were downregulated, indicating reduced developmental signaling (Figure 3b, right panel).

Old E4-O_3_ mice exhibited the upregulation of *Nefh*, *Ebf3*, and *Sfrp5* indicating cytoskeletal integrity, neuronal differentiation, and oxidative stress protection, while neurotoxic and inflammatory genes *Lcn2* and *Nr4a1* were downregulated (Figure 3d, right panel). This suggests E4 resilience to O_3_-induced neurodegeneration. Conversely, old E3-O_3_ mice showed the downregulation of *Mt2* (redox response), indicating weakened oxidative stress defense (Figure 3c, right panel), highlighting E3 vulnerability to O_3_ exposure.

To further examine the effects of O_3_ exposure on AD-related transcriptomic changes in the hippocampus, we focused on four major biological processes linked to AD pathology: oxidative stress, inflammation, neurodegeneration, and synaptic function. Older O_3_-exposed E3 mice uniquely upregulated BBB-related genes (*Hbb-bs*, *Hbb-bt*, *Hba-a1*, *Hba-2a*) (Figure 3e, green rectangles), a pattern absent in other groups, suggesting compensatory BBB repair due to heightened vulnerability. In contrast, E4 mice exhibited stronger neuroimmune regulation, with *Serpina3n*, *Zeb2*, and *Plp1* upregulated, while these same genes were downregulated in E3 mice, indicating weaker inflammatory regulation in E3. Similar opposing patterns were observed in neurogenesis and synaptic function clusters, reinforcing E4’s adaptive resilience and E3’s susceptibility to O_3_-induced neurodegeneration (Figure 3e). These findings suggest that E4 mice possess protective mechanisms against O_3_-induced neurodegeneration through enhanced antioxidant defenses, immune regulation, and neuroprotection, whereas E3 mice exhibit heightened vulnerability due to weaker oxidative stress responses, BBB dysfunction, and neuroinflammatory dysregulation. This underscores the complex interplay between ApoE genotype, aging, and environmental factors in AD risk.

### 2.3. Dissect the Molecular Mechanisms Underlying Increased Susceptibility of Old E3 Mice Exposed to O_3_

In our previous research [10], we observed significant memory impairment in old E3 male mice exposed to O_3_, whereas old E4 mice showed relative resilience, with no memory deficits. To investigate the molecular basis of this E3 vulnerability and E4 resilience, we analyzed hippocampal gene expression, using KEGG and GSEA pathway enrichment and volcano plot analyses. A total of 959 DEGs were identified between O_3_-exposed old E4 and E3 mice, with 565 upregulated and 394 downregulated in E4 (Appendix A). KEGG pathway analysis (Figure 4a) revealed neuroprotective adaptations in E4 mice, indicated by the upregulation of Hippo signaling, fatty acid elongation, cGMP-PKG signaling, and Wnt signaling. Moreover, pro-inflammatory and oxidative stress pathways such as TNF signaling and AGE-RAGE signaling were downregulated in old E4 mice, indicating reduced neuroinflammation and oxidative damage (Figure 4a). Additionally, immune-related GSEA pathways including B cell receptor signaling and FCERI-mediated NF-κB activation were also suppressed, further limiting inflammatory responses in E4 mice (Figure 4b). The list of genes of the KEGG pathway and GSEA pathway are shown in (Appendix A).

A volcano plot analysis (Figure 4c) highlights key molecular adaptations in old E4 mice that may prevent AD pathology under O_3_ exposure. The downregulation of *Grm2*, *Wnt7b*, and *Gsg1l* likely reduces excitotoxicity, enhances neurogenesis via Wnt signaling, and prevents maladaptive synaptic remodeling, supporting cognitive resilience. Meanwhile, the upregulation of *Trpv4* and *Steap1* suggests enhanced calcium homeostasis and redox balance, mitigating oxidative stress and excitotoxicity in E4 old mice. These findings suggest that E4 mice compensate for O_3_-induced neurodegeneration through enhanced synaptic plasticity, reduced neuroinflammation, and better oxidative stress regulation, highlighting genotype-dependent differences in AD risk.

A Gene Ontology (GO) analysis of old E4 vs. E3 mice exposed to O_3_ revealed key biological processes associated with neuroprotection in E4 mice but showed opposite expression patterns in old E3 mice (Figure 5a–d). In E4 mice, upregulated pathways included neurogenesis, axon ensheathment, and glial cell differentiation, suggesting enhanced neuronal growth, myelination, and structural support (Figure 5a). Conversely, these pathways were downregulated in old E3 mice, indicating impaired neuronal maintenance and myelin integrity, which may contribute to cognitive decline. Downregulated pathways related to neuron apoptotic regulation in E4 mice (Figure 5b) indicated reduced neuronal death and potential neuroprotection, while E3 mice showed no such protective regulation).

In the cellular components’ pathway, E4 mice exposed to O_3_ exhibited upregulated features like the myelin sheath and basement membrane (Figure 5c), suggesting enhanced axonal protection and extracellular matrix stabilization. Downregulated components in E4 mice, including the hemoglobin complex, neuron projection terminus, axon terminus, and presynaptic membrane (Figure 5d), suggest reduced oxidative stress and excitatory neurotransmission. In contrast, E3 mice displayed upregulation in these components, which may contribute to their heightened oxidative stress and excitotoxicity. These opposing expression patterns between E4 and E3 mice underscore genotype-specific responses to O_3_ exposure. While E4 mice exhibit protective adaptations, such responses are not observed in E3 mice, potentially explaining their heightened vulnerability to AD-related pathology.

To further understand the impact of O_3_ exposure on AD-related transcriptomic changes, we compared the gene expression patterns between old E4 and E3 mice exposed to FA or O_3_ in redox system, inflammation, neurodegeneration, synaptic function, and cellular senescence. Redox heatmaps (Figure 6a) show that old E3 O_3_-exposed mice showed an increased expression of hemoglobin subunits *Hbb-bs*, *Hba-a1*, *Hbb-bt*, and *Hba-2a*, compared with the rest groups. In contrast, E4 mice showed an upregulation of several redox-related genes (*Yme1l1*, *Vamp3*, *Prr5l*, *Ndufs1*, *Mag*, *Lpar1*, *Itgb4*, and *Tfrc*) compared with E3 mice. These findings further suggest enhanced redox regulation and antioxidant defense in E4 mice. The inflammatory heatmap shows that anti-inflammatory genes such as *Sema3e*, *Trim24*, *Zeb2*, and *Cntnap2* are upregulated in E4 vs. E3 mice (Figure 6b).

Conversely, pro-inflammatory genes such as *Cd44*, *Calca*, *Il17d*, *Pla2g3*, *Adipoq*, and *Casp1* were downregulated in O_3_-exposed E4. These findings underscore distinct, genotype-specific inflammatory pathways, with old E4 mice showing a more regulated balance of pro- and anti-inflammatory responses.

The neurogenesis heatmap (Figure 6c) reveals that E4 mice exhibit an increased expression of genes supporting neurogenesis and myelin integrity (*Fgf10*, *Mag*, *Mal*, *Myrf*, *Tspan2*), as well as genes involved in cell adhesion, neuronal survival, and differentiation (*Itgb4*, *Lpar1*). Similarly, in the synaptic function heatmap (Figure 6d), E4 mice show an upregulation of genes associated with synaptic vesicle fusion, neurotransmitter release, and vesicle transport (*Vamp3*, *Syt12*, *S1pr5*, *Rab6a*), synaptic plasticity, neuroprotection, and synaptic integrity (*Gpr37*, *Chn2*, *Tfrc*, *Actb*), as well as myelin formation (*Bcas1*) and calcium signaling (*Cacnb2*). These findings suggest that E4 mice exhibit enhanced neurogenesis, and neuronal survival compared with old E3 mice.

In the senescence heatmap (Figure 6e), old O_3_-exposed E4 mice show the upregulation of anti-senescence genes such as *Smarca5*, *Irf6*, and *Cul4b*, suggesting protective mechanisms against ozone-induced senescence. Conversely, pro-senescence genes (*Mme*, *Cd44*, *Igfbp5*, and *Map2k6*) were downregulated in E4 mice (Figure 6e). Immunostaining (Figure 6f) further reveals increased markers of cellular senescence (mH2A in astrocytes, p53 in neurons), with a more pronounced rise in old O_3_-exposed E3 mice than in old O_3_-exposed E4 mice, indicating a heightened susceptibility to O_3_-induced senescence in the former. Together, these findings highlight distinct genotype-specific mechanisms of adaptability and resilience and provide insights into the molecular basis of AD-related pathology under ozone-induced stress.

## 3. Discussion

The etiology of LOAD is complex, involving genetic predispositions, environmental factors, and age-related mechanisms. The *apoE4* allele is widely recognized as a major genetic risk factor for LOAD [7,20,21,22]. While the impact of E4 is well-established, the role of environmental factors, such as air pollution, in accelerating LOAD is less clear but increasingly relevant. Exposure to pollutants like O_3_ and nanoparticulate matter (nPM) has been linked to higher incidence and severity of AD, indicating that environmental stressors may exacerbate neurodegenerative processes [17,23,24]. Our previous work suggested that O_3_ exposure, in combination with aging, leads to memory deficits in old male E3 mice but not old E4, indicating a complex interplay between genetics, environmental exposure, and aging in AD progression [10]. Building on these findings, this study investigated the molecular mechanisms driving memory loss in O_3_-exposed old E3 male mice through the transcriptomic analysis of hippocampal tissue. Our findings demonstrate significant disruptions in BBB integrity, redox balance, neurogenesis, and synaptic function pathways in O_3_-exposed old E3 male mice, alongside an enhanced senescence signature. These results underscore the heightened vulnerability of old E3 male mice to O_3_-induced oxidative stress and neurodegeneration, supporting earlier observations of genotype-specific differences in resilience. The combined effects of activated BBB function, a dysregulated redox system, synaptic function, and impaired neurogenesis likely accelerate neurodegenerative processes, providing mechanistic insights into the molecular basis of memory decline in old E3 male mice. *ApoE* ε4 is a well-documented genetic risk factor for AD, especially impacting women [20,25,26,27]. Studies in ApoE-targeted replacement (TR) mice have shown that female ApoE4 mice tend to experience memory deficits even in the absence of environmental challenges, while male ApoE4 mice generally do not show these impairments [21,28,29]. This sex-specific vulnerability in females has led to a research focus on female responses to E4, leaving male-specific effects less understood.

Our previous and current findings reveal a new perspective, showing that the *ApoE ε4* allele may protect old male mice from O_3_-induced memory loss. This unexpected finding suggests that the E4 genotype may confer age- and sex-specific protective effects. Supporting our observations, Peris-Sampedro et al. reported that male E3, but not male E4, mice exhibited memory impairments following exposure to the pesticide chlorpyrifos, while postnatal exposure impaired memory in female E3, but not E4, mice [30,31]. These findings underscore the complex interactions between genotype, sex, and age in modulating responses to environmental stressors. Further supporting this complexity, an epidemiological study suggested that E4 carriers are more sensitive to O_3_-induced cognitive decline, although it did not address sex-specific differences, leaving the role of sex in these responses unresolved [17]. While E3 is traditionally considered a “neutral” genotype in AD-related risks, our data reveal that old E3 male mice exhibited significant reductions in antioxidant genes, indicative of a maladaptive compensatory response to oxidative damage. In addition, O_3_ exposure may exacerbate oxidative stress and neuroinflammation in old E3 mice, potentially contributing to declines in redox homeostasis markers and an upregulation of inflammatory mediators that could amplify neuronal damage and synaptic dysfunction. Conversely, old E4 mice appear to exhibit resilience, possibly through the upregulation of redox homeostasis genes, which might help counteract oxidative stress and mitigate inflammation. This resilience aligns with findings that *ApoE4* carriers maintain better vascular health under certain environmental conditions, reducing the risk of memory impairments and neurotoxicity associated with O_3_ exposure [32,33,34]. Our findings align with the existing literature demonstrating that the E3 and E4 genotypes drive distinct molecular mechanisms in AD pathology. Transcriptomic analyses of LOAD have shown that *ApoE ε3/ε4* expression primarily influences glucose and insulin signaling pathways, potentially disrupting metabolism and insulin responses [35,36]. Our findings suggest that these genotype-specific adaptive pathways may also influence responses to environmental stressors. E4’s association with enhanced antioxidant responses could explain its reduced susceptibility to O_3_-induced BBB disruption, raising the possibility that E4 may exert protective effects under certain environmental conditions. This challenges its conventional classification as the more vulnerable genotype. While E4 remains a recognized AD risk factor, particularly in women [20,25,26], our findings highlight that old male E3 carriers may be uniquely sensitive to environmental pollutants like O_3_. This sensitivity is especially relevant for individuals with occupational exposure to pollutants, as the high prevalence of E3 in the general population suggests a significant public health impact. Our results indicate that E3, traditionally viewed as “neutral”, may carry underappreciated risks, particularly in aging individuals exposed to environmental stressors like ozone, underscoring the importance of considering genotype–environment interactions in AD risk assessments.

Our study emphasizes aging as a key factor that amplifies AD progression by impairing the redox system, anti-inflammatory processes, neurogenesis, and synaptic function, while promoting cellular senescence in response to O_3_ exposure. While early molecular changes were observed in both young E3 and E4 mice, significant neurodegeneration and cognitive decline signatures were primarily confined to older E3 mice, emphasizing the role of age in heightening AD-related vulnerabilities. This finding aligns with prior studies showing that early molecular markers, such as amyloid processing, tau phosphorylation, and neuroinflammation, emerge in young models but that clinical symptoms, including memory impairment, become pronounced with age as oxidative stress and chronic inflammation accumulate [37,38,39]. For instance, Mathys et al. demonstrated that young 5xFAD mice exhibit early AD markers in the absence of clinical symptoms, which only manifest as oxidative stress and inflammation increase with age [37]. Similarly, Keren-Shaul et al. identified disease-associated microglia (DAM) in young 5xFAD mice, with significant neurodegeneration and cognitive decline becoming evident in older animals, underscoring aging’s role in exacerbating pathological changes [38].

Environmental pollutants, particularly O_3_, are increasingly recognized as accelerators of AD pathology by promoting oxidative stress, neuroinflammation, and BBB disruption. O_3_ exposure elevates oxidative stress in brain regions critical for memory and hormonal regulation, such as the hippocampus and hypothalamus, while also increasing peripheral pro-inflammatory molecules, which intensify neuroinflammation and may facilitate amyloid plaque formation [40]. Similarly, nanoparticulate matter (nPM) disrupts antioxidant defenses and promotes inflammation, with *ApoE4* carriers showing amplified inflammatory responses to nPM exposure [41,42,43]. However, our findings suggest that male E3 mice, rather than E4, are more susceptible to O_3_-induced neurodegeneration, highlighting distinct genotype-specific differences in response to pollutants such as O_3_ versus nPM. These results also point to potential sex-specific variations that warrant further investigation. Jiang et al. further demonstrated that O_3_ exposure in aging male E3 mice significantly increases oxidative stress and neuroinflammation in the hippocampus, resulting in memory impairments, while E4 mice remain resilient. This genotype-specific vulnerability aligns with evidence that pollutants like O_3_ and nPM have cumulative neurotoxic effects, particularly in genetically predisposed individuals [10,40]. Our earlier data revealed that young E3 mice initially have higher baseline glutathione (GSH) levels compared with E4 mice. However, aging and O_3_ exposure led to significant declines in GSH and cysteine (Cys) levels in E3 mice, exacerbating their vulnerability to oxidative stress. Conversely, E4 mice exhibit higher baseline antioxidant enzyme activity, with older E4 males maintaining elevated levels, likely providing compensatory protection against O_3_-induced oxidative damage. Our transcriptomic analysis supports this resilience in old E4 mice, showing the upregulation of redox homeostasis genes that could help counteract chronic oxidative stress. In contrast, old E3 mice display a less robust redox response, underscoring their heightened susceptibility to environmental pollutants and aging-related oxidative damage. These findings suggest that aging male E3 carriers, particularly those exposed to pollutants, face an elevated risk of accelerated AD pathology. This risk is further compounded by occupational exposure to pollutants such as O_3_, traffic-related air pollution, diesel exhaust, and industrial metals, which are more prevalent in industries like construction, manufacturing, and transportation, where men are disproportionately represented [44]. While *ApoE ε4*-positive AD patients are often associated with higher oxidative stress and reduced antioxidant defenses compared with non-*ApoE ε4* carriers, most studies neglect sex-specific variations [45,46,47]. By identifying the unique vulnerabilities of aging male E3 carriers, our study emphasizes the need for targeted environmental and occupational health interventions that address both genetic predisposition and pollutant exposure. Protecting high-risk populations from pollutant-induced oxidative stress could help mitigate AD risk, particularly in aging males with the *ApoE3* allele. Future research should prioritize identifying redox-sensitive proteins and molecular pathways underlying this vulnerability, providing insights for preventive strategies to reduce AD progression in susceptible groups.

Our findings reveal that BBB disruption is a critical mediator of O_3_-induced neurodegeneration in aging ApoE3 mice (Figure 7). Ozone exposure generates ROS that damage the BBB, allowing harmful substances to infiltrate the brain and triggering chronic neuroinflammation [48]. This inflammatory response, coupled with oxidative damage, disrupts neuronal signaling, accelerates lipid peroxidation, and produces toxic byproducts [49,50].

These processes impair hippocampal neurogenesis and synaptic plasticity, key components of memory formation and brain repair, ultimately contributing to cognitive decline [51,52]. In old E3 mice exposed to O_3_, BBB-related genes (*Hbb-bs*, *Hba-a1*, *Hbb-bt*, and *Hba-2a*) were upregulated, likely as a compensatory response to oxidative and inflammatory stress. However, this response appears insufficient to fully protect against O_3_-induced damage. BBB disruption can lead to glutamate dysregulation and excitotoxicity, a well-accepted phenotype in AD, and thus contribute to memory loss. Glutamate excitotoxicity follows excessive glutamate signaling, causing calcium overload, mitochondrial dysfunction, and neuronal death. In E3 mice, the combination of O_3_-induced BBB disruption and impaired calcium homeostasis (*Cacnb2* downregulation) likely exacerbates excitotoxic damage, leading to memory impairment [53,54,55]. Additionally, the upregulation of hemoglobin-associated genes in these mice may reflect an attempt to counteract oxidative and inflammatory stress but could further compromise BBB integrity. In contrast, old E4 mice exposed to O_3_ showed the downregulation of BBB-related genes but maintained memory function, suggesting stronger protective mechanisms. Despite its association with increased AD risk, ApoE ε4 may offer adaptive advantages under oxidative stress, enhancing lipid metabolism and stress responses to mitigate O_3_-induced damage [7]. The upregulation of *Cacnb2*, *Bcas1*, and *Tfrc* in E4 mice may further support resistance to excitotoxicity and preserve cognitive function. The BBB plays a critical role in maintaining brain homeostasis, particularly in glutamate clearance. Dysfunction in the BBB, as seen in AD, can lead to increased extracellular glutamate levels, neuroinflammation, and oxidative stress, further exacerbating excitotoxicity [56,57]. O_3_ exposure amplifies these effects, creating a feed-forward loop that worsens neuronal damage. In humans, chronic air pollution exposure, including O_3_, has been linked to cognitive decline and neurodegeneration, highlighting the need to examine genetic and environmental interactions in AD pathology [58]. These findings underscore the importance of targeting BBB integrity as a therapeutic strategy, particularly for high-risk populations such as aging ApoE3 carriers. Further research is required to validate these findings and explore their implications for human health. This study has several limitations. The use of ApoE-targeted replacement mice may limit the generalizability of findings to human populations. Additionally, the exclusive focus on male mice overlooks significant sex differences in AD progression, particularly in E4 carriers. Future studies should include both sexes and consider the cumulative effects of long-term pollutant exposure, as humans are typically exposed to multiple pollutants simultaneously. Investigating potential therapeutic interventions targeting antioxidant defenses, senescence, and BBB integrity could offer new strategies to mitigate AD progression in genetically predisposed individuals.

## 4. Materials and Methods

### 4.1. Animals and Exposure

Three-month-old (3M) and seventeen-month-old (17M) male human E4 and E3 gene-targeted replacement (TR) mice were obtained from Taconic and housed at the University of Alabama at Birmingham (UAB). Mice were subjected to either eight cycles of O_3_ exposure or filtered air (FA) at the UAB environmental exposure facility. Each cycle consisted of 5 days of FA/O_3_ exposure (0.8 ppm), for 7 h per day followed by 9 days of FA. A total of 14 to 16 mice were used per genotype, age, and treatment group. The O_3_ dose of 0.8 ppm was administered based on guidelines from the US Environmental Protection Agency (EPA) [15]. During the exposure, mice were provided with water but were deprived of food to prevent interference from O_3_-oxidized food components. All procedures were approved by the Institutional Animal Care and Use Committee at UAB.

### 4.2. Tissue Collection and RNA Isolation

After completion of O_3_ exposure, some mice were euthanized by isoflurane overdose followed by bilateral thoracotomy. Brains were dissected sagittally into right and left hemispheres. The right hemisphere was fixed in 10% phosphate-buffered formalin, while the left hemisphere was dissected to isolate the hippocampus and cerebral cortex, which were snap-frozen in liquid nitrogen. Total RNA was extracted from the hippocampus using TRIzol reagent (Invitrogen; Thermo Fisher Scientific, Inc., Walthan, MA, USA) [59]. RNA concentration and purity were measured using a NanoDrop ND-1000 spectrophotometer (NanoDrop Technologies, Wilmington, DE, USA), and RNA integrity was evaluated using the RNA Nano 6000 Assay Kit on the Agilent Bioanalyzer 2100 system (Agilent Technologies, Santa Clara, CA, USA).

### 4.3. Library Preparation, Sequencing, and Data Processing

A total of 1 µg RNA per sample was used for library preparation using the NEBNext^®^ Ultra™ RNA Library Prep Kit for Illumina^®^ (NEB, Ipswich, MA, USA) according to the manufacturer’s protocol. mRNA was purified using poly-T oligo-attached magnetic beads, fragmented under elevated temperature, and reverse transcribed into cDNA using random hexamer primers. The second-strand cDNA was synthesized using DNA polymerase I and RNase H, followed by end repair, adaptor ligation, and size selection (150–200 bp). Amplified libraries were purified and quality-checked using the Agilent Bioanalyzer 2100, Santa Clara, CA, USA. The sequencing libraries were clustered on the Illumina NovaSeq platform using paired-end sequencing to generate high-quality reads for each sample. Sequencing data underwent quality control using fastQC and adapter trimming with Trimmomatic. Clean reads were aligned to the mouse reference genome (GRCm38) using HISAT2. The alignment files were evaluated for quality and depth.

### 4.4. Bioinformatics and Statistical Analysis

Bulk RNA sequencing was utilized to investigate differential gene expression across various conditions. Gene expression levels were quantified using featureCounts, and normalized expression values were computed as counts per million (CPM). DESeq2 version 1.440 [60] was employed to identify DEGs across various comparisons. For the overall comparison between genotypes (E4 vs. E3) and age groups (17M vs. 3M), DEGs were identified using a threshold of *p*-value < 0.05 and absolute log2 fold change > 0.3. For exposure comparisons (O_3_ vs. FA), DEGs were identified with a cutoff of *p*-value < 0.05 and absolute log2 fold change > 0.1. Similarly, DEGs for individual group comparisons were generated using a *p*-value < 0.05 and absolute log2 fold change > 0.1. Functional analysis of DEGs was conducted using the DAVID and clusterProfiler bioinformatics tool [61,62] to identify significantly enriched pathways. This included examining KEGG pathways and GO categories, encompassing biological processes, cellular components, and molecular functions. KEGG pathway enrichment was filtered using a significance threshold of *p* < 0.05, while GO-enriched categories were considered significant with a *p*-value cutoff of 0.05 and a q-value cutoff of 0.1. To further explore the biological significance of DEGs in the comparison of group E4 vs. E3 in 17M old O_3_-exposed mice. Heatmaps were generated to visualize DEGs across functional categories. Additionally, KEGG and GSEA enrichment analyses were performed to elucidate the functional implications of the DEGs. Figures were primarily created using the R version 4.4.1 packages fgsea, pheatmap, and ggVennDiagram [63,64]. RNA-Seq raw data and metadata are available at ArrayExpress with the accession number E-MTAB-14917. Please see https://urldefense.com/v3/__http://www.ebi.ac.uk/arrayexpress/help/FAQ.html*cite__;Iw!!NoSwA-eRAg!GSp16wEjLgJoihwwly89zvXZXhZjqL5p82Qd9SGGx-2IgGhJB8iSdbc_w0yDSeiH321zx8zhGVNGF2TQhtM$ (accessed on 3 March 2025).

### 4.5. RT-qPCR Validation

To validate the RNAseq expression profiles, RT-qPCR was performed on selected genes. First, 1 µg of total RNA was reverse transcribed using the iScript Supermix for RT-qPCR (Bio-Rad Laboratories Inc., Hercules, CA, USA). RT-qPCR was conducted using iTaq Universal SYBR-Green Supermix (Bio-Rad Laboratories Inc.) on the CFX96 Real-Time PCR Detection System (Bio-Rad Laboratories Inc., Hercules, CA, USA). Primer sequences are provided in Appendix A. Relative expression levels were normalized to *Gapdh* and calculated using the 2^−ΔΔCt^ method [65]. Reactions were run in triplicate, and the data were analyzed using Bio-Rad CFX Manager software version 3.0.

### 4.6. Immunofluorescence Staining

Paraffin-embedded tissue sections (4–5 µm) were de-paraffinized in xylene (Sigma-Aldrich, The Woodlands, TX, USA) for 2 × 5 min, rehydrated in graded ethanol (100%, 95%, 70%; Sigma-Aldrich) for 3 min each, and washed in PBS with 0.05% Tween 20 (PBST, Sigma-Aldrich) for 10 min. Antigen retrieval was achieved by microwaving the sections in 0.1 M citrate buffer (pH 6.0, Sigma-Aldrich) for 5 min at 350 W, followed by cooling for 20 min. Slides were washed twice in PBST (3 min each) and incubated with 1 mg/mL sodium borohydride (Sigma-Aldrich) in PBS for 3 × 10 min, then thoroughly washed in PBS. Blocking was performed in 5% BSA (Sigma-Aldrich) and 5% goat serum (Sigma-Aldrich) in PBST for 1 h at room temperature. Primary antibodies (anti-p53, Santa Cruz Biotechnology, 2 µg/mL; GFAP, Sigma-Aldrich, 1:500; MacroH2A1.1, were diluted in PBS and incubated overnight at 4 °C. Slides were rinsed 3 × 5 min in PBST. Secondary antibodies (anti-rabbit-FL, Texa Red, Sigma-Aldrich, 1:100) were applied in 2% BSA and 500× diluted goat serum (Sigma-Aldrich) at 100 µL/slide, followed by incubation at 37 °C for 1 h. After washing three times for 5 min in PBST, nuclei were counterstained with DAPI (1:1000, Sigma-Aldrich), and sections were mounted with coverslips.

## 5. Conclusions

In conclusion, this study elucidates the molecular mechanisms underlying the interplay between aging, genotype, and environmental factors in LOAD. Key pathways associated with memory impairment, antioxidant defenses, synaptic function, inflammation, senescence, and neurogenesis were disrupted in older E3 male mice exposed to O_3_. These findings underscore the importance of genotype-environment interactions in shaping AD pathology and highlight potential therapeutic targets for mitigating disease progression.

## Figures and Tables

**Figure 1 ijms-26-02407-f001:**
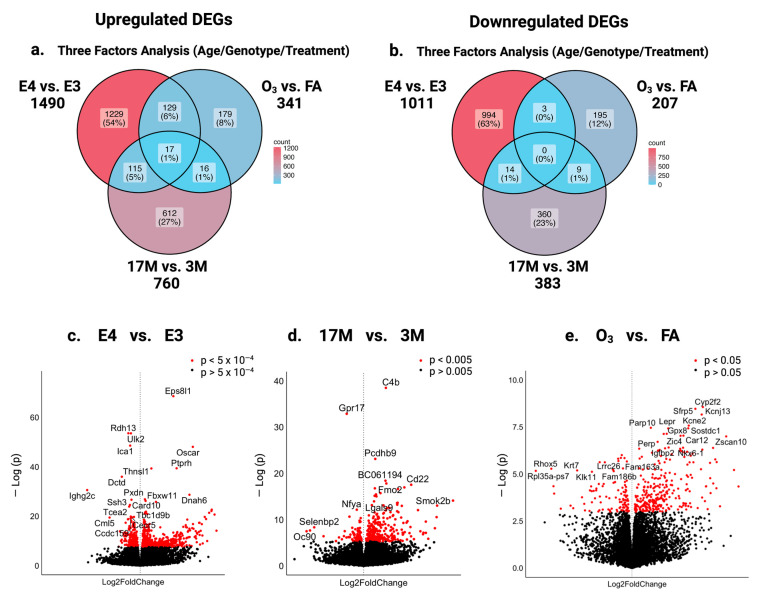
Venn diagrams show the overlap of DEGs in relation to genotype, age, and O_3_ exposure. (**a**) Upregulated DEGs and (**b**) downregulated DEGs depicted by specific factors, age, genotype, or treatment separately or in combination. (**c**,**d**) Volcano plots display DEGs for each of the three factors: (**c**) genotype; (**d**) age; and (**e**) treatment. Red dots indicate genes with *p*-values less than <0.05, while black dots represent non-significant DEGs.

**Figure 2 ijms-26-02407-f002:**
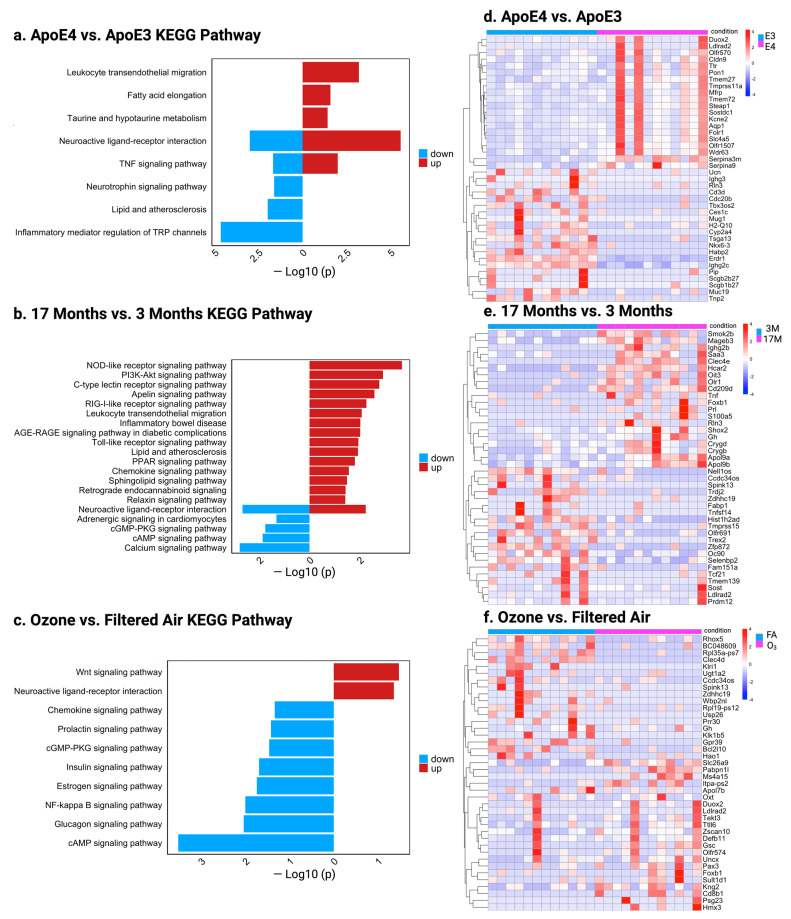
(**a**–**c**) Kyoto Encyclopedia of Genes and Genomes (KEGG) pathway analysis displays upregulated (red) and downregulated (blue) pathways for (**a**) E4 vs. E3, (**b**) 17M vs. 3M, and (**c**) O_3_ vs. FA, (**d**–**f**) Heat maps analysis of DEGs based on genotype, age, and O_3_ exposure (top 40 DEGs for each comparison): (**d**) E4 vs. E3, (**e**) 17M vs. 3M, and (**f**) O_3_ vs. Filtered Air. Red indicates upregulation, and blue indicates downregulation, with the gradient representing expression changes.

**Figure 3 ijms-26-02407-f003:**
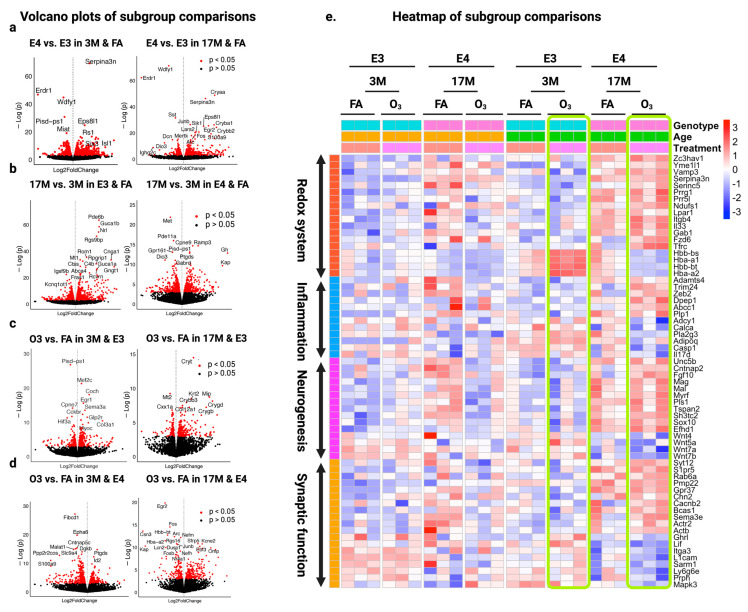
Volcano plots of DEGs in subgroup comparisons: (**a**) genotype (E4 vs. E3) at 3M (**left**) and 17M (**right**) under FA; (**b**) age (17M vs. 3M) under FA for E3 (**left**) and E4 (**right**); (**c**) treatment (O_3_ vs. FA) in E3 mice at 3M (**left**) and 17M (**right**); (**d**) treatment (O_3_ vs. FA) in E4 mice at 3M (**left**) and 17M (**right**). The heatmap (**e**) shows DEGs related to oxidative stress, inflammation, neurogenesis, and synaptic function across genotype (E3 vs. E4), age (3M vs. 17M), and treatment (FA and O_3_), with expression levels represented by color intensity ranging from −3 (downregulated) to +3 (upregulated).

**Figure 4 ijms-26-02407-f004:**
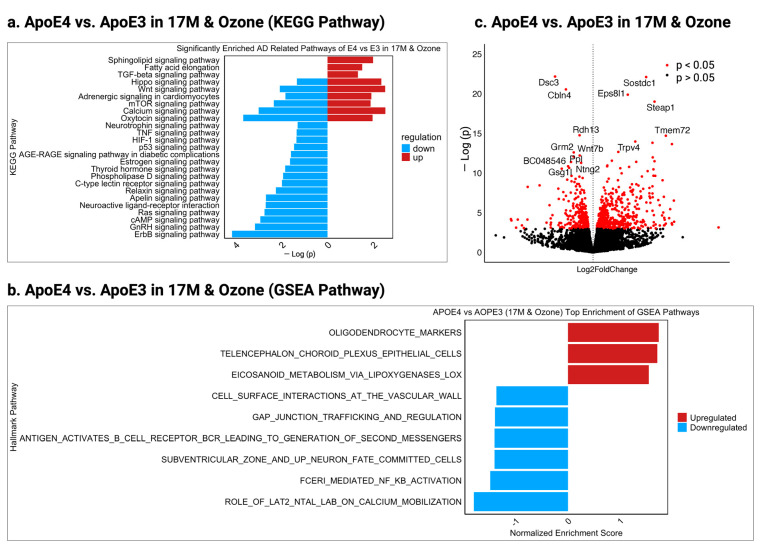
KEGG and Hallmark pathway analysis of 17M E4 vs. 17M E3 mice exposed to ozone. (**a**) KEGG pathway enrichment with upregulated (red) and downregulated (blue) pathways ranked by −log10 (*p*-value); (**b**) GSEA pathways ranked by normalized enrichment score, showing upregulated (red) and downregulated (blue) pathways; (**c**) Volcano plot of DEGs between E4 and E3, highlighting significant genes (red) with log2 fold change (x-axis) and −log10 (*p*-value) (y-axis).

**Figure 5 ijms-26-02407-f005:**
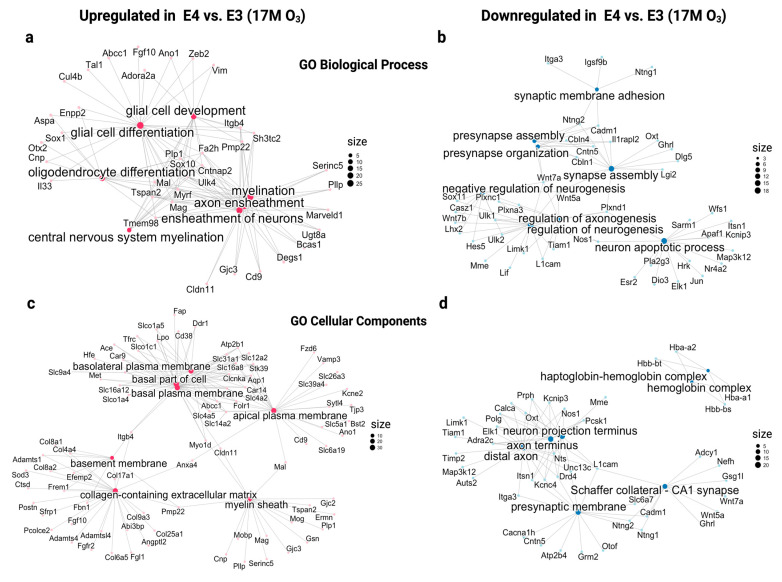
Gene Ontology (GO) enrichment network analysis for differentially expressed genes across biological processes and cellular components. Biological process for upregulated DEGs (**a**) and downregulated DEGs (**b**). Cellular components for upregulated DEGs (**c**) and downregulated DEGs (**d**). Node sizes (black) correspond to gene set sizes, with larger nodes indicating larger gene sets. Edges represent shared genes between terms. Red nodes denote upregulated genes, and blue nodes indicate downregulated genes.

**Figure 6 ijms-26-02407-f006:**
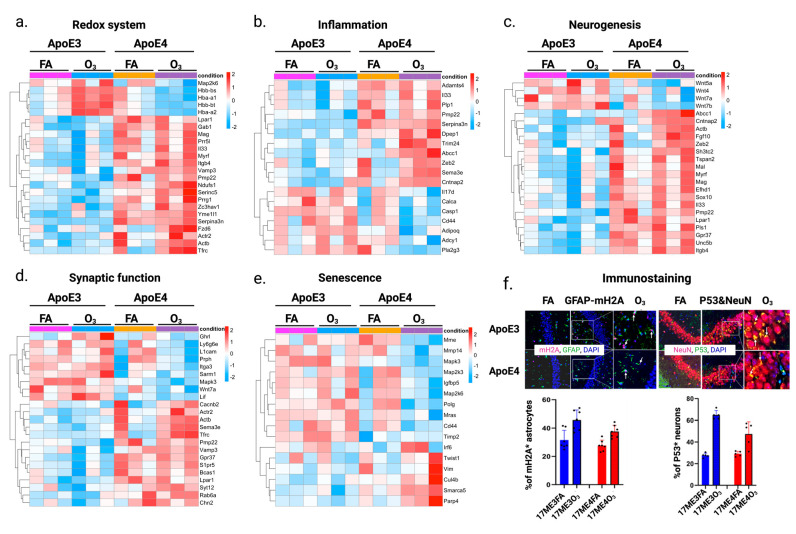
Impact of ozone exposure on oxidative stress, inflammation, neurogenesis, synaptic function, and senescence in old E4 and E3 mice. Heatmaps show changes in gene expression for (**a**) oxidative stress, (**b**) inflammation, (**c**) neurogenesis, (**d**) synapse function, and (**e**) senescence, with blue indicating downregulation and red indicating upregulation. (**f**) Immunostaining and quantification of (×20) cell senescence markers in E3 and E4 mice exposed to FA or O_3_ Green: GFAP/NeuN-are astrocyte and neuron marker respectively. Red: mH2A or p53 cell senescence markers. White arrows show mH2A positive astrocytes and p53 positive neurons. Bar graphs quantify mH2A+ astrocytes and p53+ neurons.

**Figure 7 ijms-26-02407-f007:**
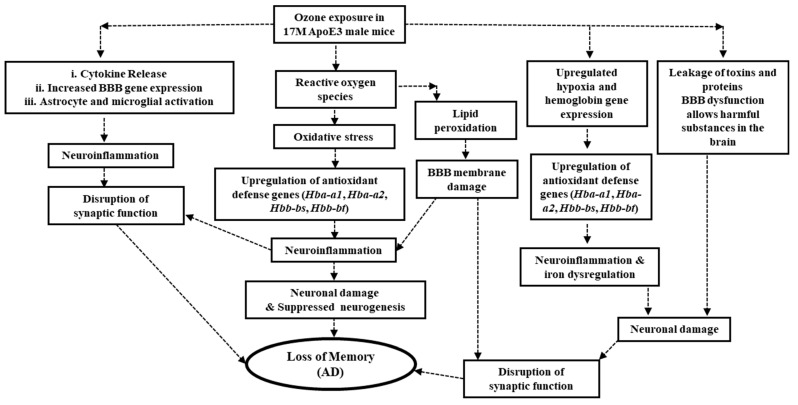
Speculative model linking ozone exposure to memory loss in 17-month-old ApoE3 male mice. This model hypothesizes that ozone exposure generates reactive oxygen species (ROS), leading to oxidative stress and lipid peroxidation, potentially damaging the blood–brain barrier (BBB) and upregulating hypoxia and hemoglobin-related genes (*Hba-a1*, *Hba-a2*, *Hbb-bs*, *Hbb-bt*). BBB dysfunction may allow the leakage of harmful substances into the brain, triggering neuroinflammation, iron dysregulation, and neuronal damage. These processes are speculated to disrupt synaptic function, suppress neurogenesis, and contribute to memory loss, characteristic of AD. This framework represents a possible mechanistic pathway, requiring further experimental validation.

## Data Availability

The original contributions presented in this study are included in the article/Appendix A. Further inquiries can be directed to the corresponding author(s).

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
