# Peer review of "Age- and ApoE Genotype-Dependent Transcriptomic Responses to O_3_ in the Hippocampus of Mice"

_ijms, 2025, doi:10.3390/ijms26062407_

Round 1

Reviewer 1 Report

Comments and Suggestions for Authors

 Manuscript title: Age- and ApoE genotype-dependent transcriptomic responses to O3 in the hippocampus of mice

 After going through the manuscript I think that it is worth publishing after minor revision.

Introduction:

-        Brief explanation of the role of ApoE polymorphism alleles as primary genetic factors influencing the risk of Alzheimer's disease should be highlighted in the introduction. i.e While the ε2 allele lowers risk, those with the ε4 allele are more likely to develop AD than those with the more prevalent ε3 variant.

-        Clarification of the mechanism through which O3 can be considered as environmental risk factor to AD should be added in the introduction. i.e the mechanism through which exposure to O3 could increase the Aβ plaque load and augmenting dystrophic neurites. (with proper and recent citation). Highlight the importance of lung-brain axis as bidirectional pathway relating exposure to O3 and AD in old individuals.

Results:

All figures should have higher resolution. The majority of them are unreadable.

Discussion:

According to the authors, upregulated genes include Cacnb2, a calcium channel subunit, Bcas1, which promotes myelin formation, and Tfrc, which is implicated in synaptic plasticity and AMPA receptor trafficking (lines 389-391). In this context, the role of glutamate excitotoxicity in Alzheimer's disease should be investigated, as well as the role of ozone as an environmental risk factor that may produce glutamate excitotoxicity as a well-accepted phenotype in AD. Of course proper citations should support this information.

Author Response

Reviewer 1
After going through the manuscript, I think that it is worth publishing after minor revision.
Introduction:
- Brief explanation of the role of ApoE polymorphism alleles as primary genetic factors 
influencing the risk of Alzheimer's disease should be highlighted in the introduction. i.e While the 
ε2 allele lowers risk, those with the ε4 allele are more likely to develop AD than those with the 
more prevalent ε3 variant.
Response: A section describing all three ApoE alleles has been inserted into the introduction and 
highlighted in red. Page 1: Lines 41-47
- Clarification of the mechanism through which O3 can be considered as environmental risk factor 
to AD should be added in the introduction. i.e the mechanism through which exposure to O3 could 
increase the Aβ plaque load and augmenting dystrophic neurites. (with proper and recent citation). 
Highlight the importance of lung-brain axis as bidirectional pathway relating exposure to O3 and 
AD in old individuals.
Response: A section detailing the mechanism of O₃'s effects on AD, along with the role of the 
lung-brain axis as a bidirectional pathway linking O₃ exposure to AD including relevant 
references has been added to the introduction and highlighted in red. Page 2: Lines 66-71.
Results:
All figures should have higher resolution. The majority of them are unreadable.
Response: all the figures have been modified and the resolution improved 
Discussion:
According to the authors, upregulated genes include Cacnb2, a calcium channel subunit, Bcas1, 
which promotes myelin formation, and Tfrc, in E4 ore significantly Ozone exposed but reduced in 
both E3 FA/o3) exposed which is implicated in synaptic plasticity and AMPA receptor trafficking 
(lines 389-391). In this context, the role of glutamate excitotoxicity in Alzheimer's disease should 
be investigated, as well as the role of ozone as an environmental risk factor that may produce 
glutamate excitotoxicity as a well-accepted phenotype in AD. Of course proper citations should 
support this information.
Response: The section on page 12 from line 427 to 461 has been modified to incorporate a 
discussion on BBB dysfunction and glutamate excitotoxicity, with additional references. All 
changes are highlighted in red.

Reviewer 2 Report

Comments and Suggestions for Authors

Nakamys and her colleagues studied the effect of O3 on age- and ApoE-dependent.

AD mice. They showed that O3 has an impact on the aged ApoE3 male mice. They identified several genes and related pathways associated with the O3-induced effects. The conducted methodologies are detailed and completed. However, the overall writing format is too tedious and should be more concise. Several minor

 comments are listed as follows.

  1. The resolution of figures and diagrams is poor.
  2. Several sections, for example, 2.1, page 4, are too tedious. The authors should extensively polish the content to make it readable for general readers.
  3. Several sentences are underlined, for example, line 159. I suppose they should be deleted.
  4. In line 423, late-onset LOAD. I guess the authors would like to say either late-onset AD or LOAD.
  5. Please put the accession number in.
Comments on the Quality of English Language

The English is fine, but the content is hard to read. 

Author Response

Reviewer 2
AD mice. They showed that O3 has an impact on the aged ApoE3 male mice. They identified 
several genes and related pathways associated with the O3-induced effects. The conducted 
methodologies are detailed and completed. However, the overall writing format is too tedious 
and should be more concise. Several minor
comments are listed as follows.
1. The resolution of figures and diagrams is poor.
Response: the resolution of the figures is modified and improved.
2. Several sections, for example, 2.1, page 4, are too tedious. The authors should 
extensively polish the content to make it readable for general readers.
Response: We have revised the whole result section thoroughly and shortened the contents in 
2.1 and other sections by 30%-40%. 
3. Several sentences are underlined, for example, line 159. I suppose they should be deleted.
Response: No sentences are underlined in the text. 
In line 423, late-onset LOAD. I guess the authors would like to say either late-onset AD or 
LOAD.
Response: “late-onset AD” has been deleted from the text from page 10: line 301
4. Please put the accession number in.
Response: The accession number will be provided a soon as we receive it

Round 2

Reviewer 2 Report

Comments and Suggestions for Authors

The authors have revised the manuscript greatly.